# Spending Money in Free-to-Play Games: Sociodemographic Characteristics, Motives, Impulsivity and Internet Gaming Disorder Specificities

**DOI:** 10.3390/ijerph192315709

**Published:** 2022-11-25

**Authors:** Jean-Michel Costes, Céline Bonnaire

**Affiliations:** 1Research Chair on Gambling, Concordia University, 2070 Mackay Street, 3rd Floor, Montreal, QC H3G 2J1, Canada; 2Laboratoire de Psychopathologie et Processus de Santé, Université Paris Cité, F-92100 Boulogne-Billancourt, France; 3“Consultation Jeunes Consommateurs”, Centre Pierre Nicole, Croix-Rouge Française, F-75005 Paris, France

**Keywords:** Free-to-Play, impulsivity, motivation, gaming experience, gaming disorder

## Abstract

Free-to-Play games (F2P) have spread widely all over the world in recent years. The current economic model for these games is based on microtransactions, where gamers can purchase additional items or services inside the game. The aim of the present study was (1) to describe the profiles and gaming patterns of F2P gamers, and (2) to compare F2P gamers who spend money and those who do not, in terms of sociodemographic characteristics, gaming experience, motivations, impulsivity, and risk of Internet gaming disorder (IGD), in a representative sample of 5062 French online gamers. Among the total sample, 68.6% were past-year F2P gamers. Among the F2P gamers, 26.1% had spent money in the game. Spending in the game was strongly associated with IGD (6.9% of F2P gamers were disordered gamers). Flow (gaming experience) and escape (motivation) were strongly associated with spending in the game and IGD. Negative urgency (impulsivity) was positively associated with spending in the game while positive urgency was positively associated with IGD. Given the strong association between spending in the game and IGD, these results highlight the importance of prevention and regulation in the field.

## 1. Introduction

Gaming has become a very popular activity for both adults and adolescents and a major leisure activity in our contemporary societies. The development of the Internet, digital technologies, and devices such as smartphones has dramatically increased in recent decades, and this increase could be associated with cases of excessive use, which often have negative health and social consequences. This booming online gaming market is experiencing very rapid changes in its economic model. Thus, the last few years have seen a shift from buying the actual games (and sometimes their monthly subscriptions) to free games (Free-to-Play, F2P) in which the player can eventually spend real world money. Known as a ‘microtransaction’, the gamers can purchase additional items, bonuses or services inside the game [1]. There are three types of microtransactions [2]: cosmetic microtransactions (e.g., aesthetic changes within the game like alternative costumes) [3], pay-to-win microtransactions—that can increase the gamer’s chances of in-game success by purchasing items or bonuses [4]—and loot boxes which provide players with a randomized reward of uncertain value [5], introducing randomness to in-game purchases [6]. As suggested by King et al. [7] (p. 10), «given the complexity and multiple demands within and across types of games, loss of control over gaming may encompass a wide range of activities, which differ according to game type». One common feature of all F2P games is the proposal of microtransactions to the player. Thus, it is important to investigate factors associated with this in-game activity and whether it is associated with loss of control over gaming (i.e., Internet Gaming Disorder).

### 1.1. Motives Associated with Microtransactions in Free-to-Play Games

Several studies have investigated motivation to purchase in-game content. For example, Hamari et al. [8] found that the purchase motivations of unobstructed play, social interaction, and economic rationale were positively associated with the amount of money spent in the game. More recently, Marder et al. [3] found that the motivation for spending in the game League of Legends was not necessarily linked to the value of the item purchased but sometimes actually aimed to support the video game developer, highlighting the importance of the act of purchasing itself. Beyond the motivation to spend money in the game, it is important to study the psychological motives for playing video games and see if there are specific motives associated with specific video game types. Indeed, understanding the specific factors that lead an individual to engage in gaming and in specific types of video game will shed light on the mechanisms of healthy, excessive, and disordered gaming. Several studies showed that gaming motives is an important risk factors of IGD [9,10,11]. To our knowledge, no study has investigated motivations associated specifically to F2P and those associated with in-game purchasing. Furthermore, analyzing gameplay and player experiences sheds light on players’ game choices [12] and contributes to determining gamers’ motives. Little is known about the player experience in F2P games and whether the player experience differs between gamers who play F2P without spending money and those who spend money in the game.

### 1.2. Microtransactions in Free-to-Play Games and Its Association with Internet Gaming Disorder

In their study about the exposure to the three types of microtransactions in the most popular Steam games between 2010 and 2019, Zendle et al. [2] concluded that cosmetic microtransactions and loot boxes are present in games played by the majority of desktop gamers within their sample: over 80% of gamers had played a game with cosmetic microtransactions and over 70% with loot boxes in 2019. Yet, academics and mental health workers have expressed concern about the detrimental effects of the incorporation of microtransactions in modern video games. These ‘predatory monetization schemes’ [13] not only create potential financial damages [14], but also increase the risk of developing Internet gaming disorder (IGD) [15,16] since F2P games encourage casual players to become more hardcore gamers [17]. Dreier et al. [18] found that 5.2% of their sample of F2P gamers met criteria for IGD. They also found a relationship between higher amounts of money spent on F2P games and IGD. However, only few studies have investigated the relationship between IGD and microtransactions [16,19,20,21,22]. Given the association between microtransactions, gaming-related financial harm and IGD [23], more study are needed to better understand this association. Indeed, no study have investigated if factors associated with microtransactions differ from those associated with IGD.

### 1.3. Impulsivity, Free-to-Play Games and Internet Gaming Disorder

Impulsivity has been associated with many psychopathological disorders including addictive behaviors and more specifically IGD. Several studies have found a relationship between IGD and poor self-control, sensation seeking traits, and diminished inhibitory control [24,25]. Based on both self-reported and neuroimaging measures, IGD seemed associated with heightened impulsivity [26,27,28,29]. For example, individuals with IGD have been shown to present significantly higher levels of motor and attentional impulsivity, compared to controls [30]. In a 2-year longitudinal research study, impulsivity was identified as a risk factor for developing subsequent problematic video game use [31]. Nevertheless, only a few studies have investigated the relationship between impulsivity and gaming, and except for Multiplayer Online Battle Arena [32], no study has evaluated the relationship between impulsivity and F2P, and more specifically microtransactions in F2P.

### 1.4. Current Study

Previous studies have mostly investigated the different types of microtransactions, motives to spend money inside the games and the relationship between amounts of money spent on F2P games and IGD. Nevertheless, no study has compared F2P gamers who spend money and those who do not do it inside the game. Yet, spending money inside the game, whatever the reasons, is not a trivial act and is not the behavior of all F2P gamers. Thus, profile of F2P gamers could differ in regards of spending money or not. As previous studies focused on loot box purchase, it seems important to investigate more broadly the fact of spending money in the game. Furthermore, few studies have investigated psychological factors associated with in-game spending. Given the association between impulsivity and IGD, it seems likely that individuals with traits of impulsivity would be more prone to spend money within games. Thus, it is important to better understand who these spending gamers are, why they spend money on F2P games (motives and gaming experience) and if factors associated with spending money in the game are similar to those associated with IGD.

Thus, the aim of the present study is (1) to draw up the profile of F2P gamers and describe their gaming behavior through the comparison of F2P gamers and non-F2P gamers and (2) to compare F2P gamers who spend money and those who do not, in terms of sociodemographic characteristics, gaming experience, motivations, impulsivity, and risk of IGD, and to (3) investigate if factors associated with spending money are similar to those associated with IGD, in a representative sample of French online gamers.

## 2. Materials and Methods

### 2.1. Participants

Data were derived from the 2019 eGames-France survey carried out by the French Monitoring Center on Gambling and conducted by Médiamétrie, a French private institute, specializing in web panel surveys, between the 2 and the 20 December 2019. The present study was conducted in accordance with the 1964 Helsinki Declaration and its later amendments or comparable ethical standards. All participants received information regarding the survey and all participants provided written informed consent to participate. A sample of 5062 Internet users aged 18 to 65 years old was recruited from a vast web panel database. Because quota targets have not yet been completely achieved, data were weighted and adjusted to represent the French Internet user population according to age, gender, and major socio-professional categories. Participants completed a self-administered online survey on gaming and gaming practices using a computer-assisted web interviewing technique (CAWI). Among the total sample, 3472 gamers (68.6%) were past-year F2P gamers (51.5% F2P gamers were men; mean age 39.5 years). Among F2P gamers, 26.1% (17.9% of the overall sample) had spent money in the game.

### 2.2. Measures

Participants’ sociodemographic characteristics, including gender, age, level of education, household income, and socioeconomic status were evaluated.

#### 2.2.1. Game-Play Patterns

Were included in the “Free-to-Play gamers” group all individuals stating they had played online F2P games on a laptop, tablet, or smart-phone, or on a social network, in the past 12 months. The frequency of play during the week and the weekend, the time spent doing this activity and the age they first played F2P games were evaluated.

Respondents were asked if they spent money during their gaming activity, the reason why (to significantly increase their chances to win, to gain time in the game or to keep the game going, for aesthetic reasons, or to support the gaming community), and the amount of their expenditure.

Were included in the “Spending F2P gamers” group all gamers stating that they had spent money in these games in the past 12 months.

#### 2.2.2. Gaming Experience and Motivations

Gaming experience was described using the Game Experience Questionnaire, In-game module (GEQ) by IJsselsteijn et al. [33,34]. This 14 items scale assesses game experience as scores on seven components: Sensory and imaginative immersion (i.e., aspects of how strongly connected with the game gamers felt), Flow (i.e., whether gamers lost track of their own effort and or the passage of time during the game), Competence (i.e., how well players judged their own performance against the game’s goals), Positive and Negative Affect (i.e., positive and negative emotional experiences), Tension (items related to these specific negative emotions), and Challenge (i.e., the degree to which gamers found the game to be difficult or challenging). For each component, the scores can range from 0 to 4.

The motivations for gaming were described using the Motives for Online Gaming Questionnaire (MOGQ) by Demetrovics et al. [35]. This 27-items scale measures seven dimensions of gaming motives: Escape (escaping from reality), Coping (coping with stress and distress), Fantasy (in-game identities and experience), Skill Development (such as attention and coordination), Recreation (entertainment and enjoyment), Competition (challenging and competing with others), and Social (building and maintaining social relationships). For each dimension, the scores can range from 1 to 5, except for Recreation (1 to 4). The MOGQ measures the motives of online gamers from a broader age range. It covers the major gaming motives identified in previous research and shows high internal consistency.

#### 2.2.3. Internet Gaming Disorder

The Internet Gaming Disorder Scale–Short-Form (IGDS9-SF, Pontes & Griffiths [36]) is a short psychometric tool adapted from the nine core criteria that define IGD according to the DSM-5 [37]. The aim of this instrument is to assess the severity of IGD occurring over a 12-month period. In order to differentiate disordered gamers from non-disordered gamers, participants who answered ‘5: Very often’ on at least five of the nine criteria were considered disordered gamers.

#### 2.2.4. Impulsivity

The short UPPS-P Impulsive Behavior Scale (UPPS-P; Billieux et al. [38]) is a self-report questionnaire that assess five dimensions underlying impulsive behaviour through 20 items. For each dimension, the scores can range from 1 to 4. The negative urgency and positive urgency subscales measure the tendency to act rashly when experiencing intense emotional states (either negative or positive). Lack of premeditation corresponds to difficulty in planning the consequences of one’s behaviour. Lack of perseverance corresponds to difficulty in focusing on a boring or difficult task. Finally, the sensation-seeking subscale measures the proneness to seek new and exciting.

### 2.3. Statistical Analysis

A bivariate analysis was used to compare: (1) the socio-demographic profiles of F2P-gamers and non-F2P-gamers; (2) among F2P-gamers, the socio-demographic profiles, gaming practices, motivations and impulsivity of those who spend money in game and those who do not; (3) among F2P-gamers, the socio-demographic profiles, gaming practices, motivations and impulsivity of those with a disordered gaming pattern and those without. The differences were tested for significance.

Then, three different multivariate logistic regression models were used to assess the effects of a set of predictors on three outcome variables:(1)“Free-to-Play gaming” with socio-demographic factors as dependent variables;(2)“Spending in game” with dependent variables belonging to 5 groups of predictors: Sociodemographic, Gaming patterns, Gaming experience, Gaming motives, and Impulsivity(3)“IGD” with dependent variables belonging to 5 groups of predictors: Sociodemographic, Gaming patterns, Gaming experience, Gaming motives, and Impulsivity

For each outcome variable, regressions adjusted on gender, age and all the variables belonging to the same group of predictors were calculated.

For Gaming patterns, Gaming experience and Gaming motives variables, a preliminary analysis on the correlations was carried out. The correlations between modalities for these three variables being strong, the risk of multicollinearity was assessed by calculating VIFs. This analysis led us to remove two items from the motivation variable: “Social” and “Fantasy”.

R version 4.0.2 software was used for the analysis.

## 3. Results

### 3.1. Free-to-Play Gamers

Compared to non-gamers, F2P-gamers were more frequently men, young people, from wealthier social or cultural backgrounds and with a professional activity. The multivariate regression model with socio-demographic factors shows that men, youngest people and those with an above-average level of household income were more likely to play Free-To-Play games than women, oldest people and those with an below-average level of household income. Conversely, inactive people were less likely to play these games than those with a professional activity or students. However, these socio-demographic differences remain modest, underlining that the practice of these games has spread very widely among the entire population, in all social backgrounds (see Table 1).

### 3.2. Spending Money in the Game

Among Free-to-Play gamers, spending money was more very significantly more frequent among men and the youngest participants (see Table 2).

Starting gaming recently, high gaming frequency, time spent gaming during the week-end, and playing with other gamers were associated with spending in the game.

“Flow” and “positive” or “negative affect” game experiences were positively associated with spending money. The motives “escape” and “coping” were positively associated with spending money, while “challenge”, “skill development”, and “recreation” motives were negatively associated with spending money in the game. With regard to impulsivity, “negative urgency”, “lack of perseverance”, “lack of premeditation” and “sensation seeking” was positively associated with spending in the game.

### 3.3. Internet Gaming Disorder

According to IGDS9-SF, 6.9% (n = 240) of the sample of Free-to-Play gamers were classified with gambling disorder. While the IGDS9-SF have demonstrated adequate internal consistency, excellent criterion validity, and the ability to distinguish different subgroups [39], some item are not clinically relevant and may pathologize non-problematic patterns of gaming [40].

F2P-gamers with a disordered gaming pattern compared at those without were more likely to be men and youngest people, and less likely to be people with a high or a median level of education and students (see Table 3).

Regarding playing patterns, recency of gaming, very high gaming frequency and long playing time during the week were associated with disordered gaming. Spending in the game was one of the strongest predictors of disordered gaming.

Disordered gamers had a more intense experience of “flow”, “competence”, “positive” and “negative affect”, and “challenge”. Disordered gamers had greater levels of “escape” and “coping”, and lower levels of “competition” motivations.

“Negative urgency”, “lack of perseverance”, “lack of premeditation” and “sensation seeking” impulsivity facets were positively associated with disordered gaming.

Table 4 looks in detail at the reasons for these payments and the sums involved and showed that “To gain time in the game” or “To support the community of the game” were the two reasons linked to disordered gaming. The gamers who reported spending “to gain time in the game” also spent the highest amounts. There was a strong relationship between the amount spent and disordered gaming.

## 4. Discussion

This study provided a first description of French F2P gamers and those who spend money in F2P games based on a nationwide representative survey. Playing F2P games was not very strongly associated with a specific gender, and was over represented by high socio-demographic categories and individuals under the age of 50 and mostly for those aged 30 years or less. These results are in line with data from the video game industry, which estimates the average age of video game players to be around 40 years old. This shows that gaming is much more extensive than the adolescent or young adult population alone [41,42,43]. Both genders play F2P games, although they probably play different types of F2P games [42,44,45,46]. On the contrary, spending money in the game was positively associated with being male, a student, and being between 18 and 29 years old and as in previous studies, these characteristics were also associated with IGD [47,48]. Our results also showed that spending money was significantly more common for gamers when they had recently started gaming, during the first two years and this factor is more strongly associated with spending money than with IGD. Thus, spending money could be link to the novelty of a game and this behavior could decrease during time. Longitudinal studies are necessary to confirm this hypothesis.

Gaming time was associated with both spending money in the game and IGD but for in-game spending, this refers to time spent playing during weekends (more than 5 h) whereas for IGD, it refers to time spent playing during the week (more than five hours). Thus, gaming time duration could be linked to spending money but as a leisure activity while gaming time duration is linked to IGD and it takes precedence over daily life obligations such as sleep time or academic/professional activities.

As in previous studies on loot box expenditure [49], microtransactions were a major factor associated with IGD. As in Dreier et al.’s study [18], the amount of money spent in F2P games was associated with IGD. Nevertheless, in our study, only spending more than 20 euros per month was associated with IGD and the strongest association was found in those who spent more than 50 euros per month. Monetization strategies in F2P games increase the gamer’s commitment towards the game and therefore increase the risk for disordered use of gaming [18]. Indeed, creating commitment is associated with IGD [50,51]. Thus, it is not surprising that in our study, one significant reason for spending money was to gain time in the game. Spending money allows gamers to stay in the game for longer, which has detrimental consequences on the gamers’ lives [52]. Thus, microtransactions can be seen as an important factor that mediate the relationship between time spent in the game and IGD. This result respond to King et al.’s question [7] that IGD criteria should include excessive spending in games.

Motivations associated with microtransactions and with IGD were partially similar: the desire to escape from reality and avoid real world problems, and the desire to cope with distress and enhance mood. Thus, the more the player wanted to escape, cope with distress and live a different life, the more the player spent money to stay in the game longer and the more a disordered relationship was created with the game. Furthermore, as in previous studies on IGD [53,54,55,56,57], the escape motive was the strongest predictor for this disordered relationship, especially for IGD F2P gamers. These results are congruent with those regarding gaming experience. For both microtransactions and for IGD, associations with experience of flow, and with positive and negative affect were significant. Indeed, exciting gaming experiences and losing track of effort and time were associated with spending money and of course with IGD. As previously highlighted by Griffiths [58] regarding Internet gambling, immersion and dissociation are important factors that make online activities potentially seductive and/or addictive. In one problematic cluster identified by Billeux et al. [59], they play to “dissociate” from real life more than to actually succeed in the game [60].

Experiencing specific negative emotions during the game was associated with microtransactions. Thus, spending money could sometimes be a coping strategy to deal with negative emotions generated by the game. In their study, Dreier et al. [18] found that solving problematic in-game situations by spending money might be linked to the coping mechanisms of vulnerable F2P gamers. This is an example of a maladaptive coping strategy used to deal with a negative affective state: financial investment and spending as a reaction to a problematic situation. On the other hand, competence (i.e., how well players judged their own performance) and challenge (i.e., the degree to which gamers found the game to be difficult or challenging) were associated only with IGD (not microtransactions). Some game types (for example MOBA or FPS) are highly competitive and challenging [12,61]. In their qualitative study, Johnson et al. [12] showed that competition and performance were what MOBA gamers preferred in their gaming experience. Thus, feeling competent and challenged during the game, along with the competitive aspect, are factors associated with IGD in F2P gamers.

With regard to motivations for gaming, competition, skill development, and recreation were negatively associated with microtransactions. Thus, the gamer’s desire to improve his skills and to compete with and defeat other players in order to experience a sense of achievement did not involve spending money to make progress and win. Players may rely on their own skills to do so. Likewise, the need to achieve relaxation through gaming does not imply spending money. As highlighted by King et al. [7], contrary to gambling which is a risk taking activity through betting, gaming is largely unrelated to winning or spending money.

Our results are in line with previous studies showing that gamers with high levels of recreation and skill development motives for playing [53,62,63] are less likely to present IGD in comparison to gamers with other motives [64]. However, contrary to previous results [62,64,65], in our study the desire to compete with and defeat other players in order to experience a sense of achievement (i.e., competition) was negatively associated with IGD. In the studies cited, spending money in the game was not controlled for, yet this specific characteristic of F2P appeared as a predominant factor.

Regarding impulsivity, our results showed that this personality trait is associated with both spending money in game and IGD. While the vast majority of studies find a relationship between IGD and impulsivity, very few studies on gaming have used the UPPS model (see Şalvarlı & Griffiths [66] for a recent review). Furthermore, as demonstrated by Billieux et al. [59], levels of impulsivity and its various characteristics depend on type of disordered gamers. Our results showed that F2P gamers with IGD and those who spend money inside the game had high impulsivity scores except for positive urgency... This personality trait (especially sensation seeking and lack of perseverance) could explain why these players prefer a gaming environment that is particularly stimulating, fast, and changeable, particularly in the action game genre [67]. A recent psychometric work on both clinical and non-clinical samples showed that positive and negative urgency form a single and coherent construct, meaning that differentiating these two subscales as separate constructs is not necessary [68]. Our results showed that for some IGD, investigating separately these constructs is important. The literature on gambling showed that individuals with gambling disorder are characterized by abnormal responses to pleasant stimuli and a proneness to act rashly in response to positive emotions [69]. It seems like this result is not true for F2P gamers with IGD. Further study comparing gamers and gamblers could confirm this hypothesis.

This study had limitations. First, self-report measures were used which can lead to biases (e.g., recall, social desirability, etc.). The method of recruitment used in this study (a sample of F2P gamers among a larger panel) was another limitation because it cannot guarantee perfect representativeness of the sample, though attempts were made to do so. The generalization of these results to all F2P gamers can only be made with caution. Due to the cross-sectional design, our analyses cannot imply causation between the variables. Further longitudinal studies would be needed in order to conclude about directionality between variables. One shortcoming of the present study is that it did not differentiate between different types of F2P games. On the one hand, it was interesting to have a global vision of all F2P games but on the other hand, the scope of these games is very wide and covers very different realities, in terms of audience, motives, experiences, patterns and problems encountered. Unfortunately, a relevant typology of game categories that would be relevant to structure our analyses on F2P phenomenon is lacking.

## 5. Conclusions

Our study showed that the practice of F2P is widespread in the general population, in men and women, that a large proportion of gamers spend money inside these games, and that this behavior (spending more than 20 euros per month) is strongly associated with IGD. The existence of this link, seemed to be a reasonable hypothesis, had not yet been clearly established. Indeed, most of the studies that have studied this issue have focused on a very particular form of spending, the practice of loot-boxes [16,19,20,21,22]. As far as we know, only one study has established and measured a link between spending money in Free-to-Play games and IGD, but this one was based on a sample of young people (12–18 years old) [18]. Our results allow us to consolidate the hypothesis that spending on game would be a predictive factor of gaming disorder.

The more the gamers want to escape from reality, to cope with distress and to live a different life, the more they spend money inside the game to stay longer and the more they develop a disordered relationship with the game. On the other hand, the desire to compete with and defeat other players in order to experience a sense of achievement or to play for entertainment and enjoyment leads to less in-game spending and disordered gaming. Beyond the potential detrimental association find in the literature between loot box purchase, IGD and problem gambling [49], our results showed that spending a certain amount of money in F2P games was related to disordered gaming. Indeed, Carey et al. [23] found that financial harms might occur in gaming activities that facilitate continuous spending options.

These results raise the question of prevention and regulation of this field. The difference in the level of regulation between gambling (strong) and gaming (weak) is a problem, as it may encourage gambling operators to prefer to develop game mechanisms (such as loot boxes) in the less restrictive field of gaming. From the point of view of public policy, a reflection should be conducted to explore the limits between gambling and gaming, which would make it possible to redefine the criteria that define that a game falls within the scope of gambling. This has already begun in some countries, which have banned loot boxes, considering them to be gambling. The next step would be to raise the level of regulation of solicitations to spend money on free-to plat games. Thus, harm-reduction tools regarding spending should probably be necessary, especially amongst vulnerable gamers, mostly the youngest. Thus, in addition to inform gamers of the presence of these microtransactions and their potential harm, a spending limit could be proposed by the game to prevent financial harm and continued play. Of course, these actions have little chance of being implemented by game operators whose economic model is based on these on-game spending and who have, for the moment, no legal obligations in this area but one could imagine a negotiation with them to make them aware of the need to address the problem by promoting guidelines for good practices and thus avoiding more severe regulation of the field of gaming. The limited research knowledge currently available on F2P sets the stage for future research to build on these findings.

## Figures and Tables

**Table 1 ijerph-19-15709-t001:** Free-to-Play gaming—multifactorial sociodemographic predictive model.

Population Studied: All Respondents	Bivariate Analysis			Multivariate Analysis	
	No F2P Gaming	F2P Gaming	Pearson’s X^2^ Test *p* Value	Adjusted OR	*p* Value
*N = 5062*	*1590*	*3472*			
**Sociodemographic**	(col %)				
Women	57.0	48.5		ref.	
Men	43.0	51.5	<0.001	1.37	*<0.001*
Age_18–29 (ref. 50 and more)	15.4	28.1		3.12	*<0.001*
Age_30–49	35.1	45.0		2.30	*<0.001*
Age_50 and more	49.4	26.9	<0.001	ref.	
Education ≤ Bac	26.8	20.2		ref.	
Education Bac ou+2 (ref.)	43.8	46.7		1.06	*0.486*
Education Bac+3 ou+	29.4	33.1	<0.001	0.94	*0.541*
Occupation_soc-prof. cat. inf.	36.2	37.8		ref.	
Occupation_soc-prof. cat. sup. (ref. spc inf.)	34.1	37.1		0.93	*0.397*
Occupation_students	4.4	9.0		1.14	*0.419*
Occupation_other inactifs	25.2	16.0	<0.001	0.82	*0.024*
Household income < 2500E	47.5	42.7		ref.	
Household income ≥ 2500E (ref. < 2500E)	43.0	48.6		1.27	*0.001*
Household income no answer	9.4	8.7	0.001	1.03	*0.817*

**Table 2 ijerph-19-15709-t002:** Spending in game—multifactorial predictive models.

Population Studied: Free-to-Play Gamers	Bivariate Analysis			Multivariate Analysis	
	No Spending in Game	Spending in Game	Pearson’s X^2^ Test *p* Value	Adjusted OR	*p* Value
*N = 3472*	*2567*	*905*			
**Socio-demography**	(col %)				
Women	53.1	35.5		ref.	
Men	46.9	64.5	<0.001	2.01	*<0.001*
Age_18–29	23.0	42.5		6.68	*<0.001*
Age_30–49	44.5	46.5		3.09	*<0.001*
Age_50 and more	32.5	11.0	<0.001	ref.	
Education ≤ Bac	21.3	17.1		ref.	
Education Bac ou+2	47.2	45.5		0.95	*0.642*
Education Bac+3 ou+	31.5	37.5	0.002	0.93	*0.619*
Occupation_soc-prof. cat. inf.	38.1	36.8		ref.	
Occupation_soc-prof. cat. sup.	35.3	42.4		1.15	*0.185*
Occupation_students	9.1	9.0		0.50	*<0.001*
Occupation_other inactifs	17.5	11.8	<0.001	1.04	*0.780*
Household income < 2500E	42.0	44.8		ref.	
Household income ≥ 2500E	47.9	50.6		0.91	*0.295*
Household income no answer	10.2	4.6	<0.001	0.43	*<0.001*
**Gaming patterns**	(col %)				
Start gaming 3 years or more	84.6	53.0		ref.	
Start gaming 1 or 2 years	8.5	11.1		2.35	*<0.001*
Start gaming < 1 year	6.9	35.9	<0.001	6.67	*<0.001*
Gaming freq. < 1 time by week	17.8	6.8		ref.	
Gaming freq. 1/2 time by week	21.4	13.7		1.61	*0.011*
Gaming freq. almost every day	31.4	27.8		2.39	*<0.001*
Gaming freq. many times by day	29.5	51.7	<0.001	3.99	*<0.001*
Playing time_week ≤ 5 hours	89.5	76.3		ref.	
Playing time_week > 5 hours	10.5	23.7	<0.001	1.31	*0.053*
Playing time_week-end ≤ 5 hours	91.9	71.2		ref.	
Playing time_week-end > 5 hours	8.1	28.8	<0.001	2.13	*<0.001*
Play alone	80.7	51.5		ref.	
Play together same place	9.2	29.0		3.18	*<0.001*
Play together online	10.1	19.5	<0.001	2.12	*<0.001*
**Gaming experience (GEQ score)**	(mean score)				
Sensory	1.27	1.94	<0.001	1.04	*0.498*
Competence	2.09	2.47	<0.001	1.05	*0.464*
Flow	0.44	1.29	<0.001	1.93	*<0.001*
Tension	1.20	1.83	<0.001	1.16	*0.046*
Challenge	1.80	2.21	<0.001	0.96	*0.597*
Negative Affect	1.04	1.76	<0.001	1.15	*0.028*
Positive Affect	1.52	2.21	<0.001	1.44	*<0.001*
**Motivations (MOGQ score)**	(mean score)				
Escape	1.79	2.81	<0.001	2.21	*<0.001*
Competition	2.38	3.10	<0.001	0.80	*0.027*
Coping	1.72	2.85	<0.001	2.30	*<0.001*
Skill Development	2.44	3.09	<0.001	0.80	*0.025*
Recreation	2.67	3.16	<0.001	0.75	*<0.001*
**Impulsivity (UPPS score)**	(mean score)				
Negative Urgency	2.20	2.56	<0.001	1.48	*<0.001*
Positive Urgency	2.44	2.70	<0.001	1.07	*0.485*
Lack of Premeditation	1.82	2.05	<0.001	1.49	*<0.001*
Lack of Perseverance	1.77	2.04	<0.001	1.79	*<0.001*
Sensation Seeking	2.36	2.71	<0.001	2.03	*<0.001*

**Table 3 ijerph-19-15709-t003:** Internet gaming disorder—multifactorial predictive models.

Population Studied: Free-to-Play Gamers	Bivariate Analysis			Multivariate Analysis	
	No Disorder Gaming	Disorder Gaming	Pearson’s X^2^ Test *p* Value	Adjusted OR	*p* Value
*N = 3472*	*3232*	*240*			
**Socio-demography**	(col %)				
Women	49.5	35.7		ref.	
Men	50.5	64.3	<0.001	1.61	0.001
Age_18–29	26.7	46.4		7.33	*<0.001*
Age_30–49	45.0	45.1		3.39	*<0.001*
Age_50 and more	28.3	8.5	<0.001	ref.	
Education ≤ Bac	19.9	24.2		ref.	
Education Bac ou+2(ref.)	47.5	36.1		0.45	*<0.001*
Education Bac+3 ou+	32.6	39.7	0.004	0.58	*0.010*
Occupation_soc-prof. cat. inf.	37.5	41.3		ref.	
Occupation_soc-prof. cat. sup. (ref. spc inf.)	36.7	43.0		1.04	*0.841*
Occupation_students	9.2	6.4		0.39	*0.002*
Occupation_other inactifs	16.5	9.3	0.009	0.61	*0.063*
Household income < 2500E	42.6	44.6		ref.	
Household income ≥ 2500E (ref. <2500E)	48.3	52.1		1.02	*0.915*
Household income no answer	9.1	3.3	0.009	0.39	*0.012*
**Gaming patterns**	(col %)				
Start gaming 3 years or more	79.3	36.7		ref.	
Start gaming 1 or 2 years	9.1	9.8		1.83	0.019
Start gaming < 1 year	11.6	53.5	<0.001	3.97	*<0.001*
Gaming freq. < 1 time by week	15.7	5.0		ref.	
Gaming freq. 1/2 time by week	19.9	12.2		1.19	0.644
Gaming freq. almost every day	31.4	18.0		0.98	0.964
Gaming freq. many times by day	33.1	64.8	<0.001	2.41	0.007
Playing time_week ≤ 5 hours	87.3	69.0		ref.	
Playing time_week > 5 hours	12.7	31.0	<0.001	1.72	0.005
Playing time_week-end ≤ 5 hours	88.0	66.8		ref.	
Playing time_week-end > 5 hours	12.0	33.2	<0.001	1.15	0.445
Play alone	75.0	47.3		ref.	
Play together same place	12.7	37.1		1.49	0.032
Play together online	12.3	15.6	<0.001	1.02	0.922
Spending money in game—No	78.1	18.3		ref.	
Spending money in game—Yes	21.9	81.7	<0.001	6.15	*<0.001*
**Gaming experience (GEQ score)**	(mean score)				
Sensory	1.36	2.56	<0.001	1.05	0.663
Competence	2.14	2.93	<0.001	1.30	0.042
Flow	0.56	2.05	<0.001	2.30	<0.001
Tension	1.29	2.38	<0.001	1.10	0.438
Challenge	1.84	2.76	<0.001	1.34	0.016
Negative Affect	1.14	2.44	<0.001	1.48	0.001
Positive Affect	1.62	2.80	<0.001	1.68	<0.001
**Motivations (MOGQ score)**	(mean score)				
Escape	1.93	3.70	<0.001	3.95	<0.001
Competition	2.48	3.74	<0.001	0.55	0.001
Coping	1.89	3.68	<0.001	2.50	<0.001
Skill Development	2.52	3.77	<0.001	1.04	0.846
Recreation	2.72	3.80	<0.001	0.98	0.882
**Impulsivity (UPPS score)**	(mean score)				
Negative Urgency	2.25	2.87	<0.001	2.63	<0.001
Positive Urgency	2.48	2.89	<0.001	0.98	0.888
Lack of Premeditation	1.87	2.06	<0.001	1.57	0.003
Lack of Perseverance	1.83	2.04	<0.001	1.94	<0.001
Sensation Seeking	2.42	2.93	<0.001	2.90	<0.001

**Table 4 ijerph-19-15709-t004:** Spending in game—reasons for payments and amounts spent.

	% Among Spending Gamers	Average Monthly Spending	% Disorder Gaming	OR *	*p* Value
**Reasons of payment**					
To increase chances to win	28.1	197€	24.3	1.09	*0.652*
To win time in the game	21.7	174€	30.1	1.60	*0.021*
To keep the game going	34.4	50€	25.3	1.42	*0.059*
To enjoy the game	36.4	128€	20.9	1.04	*0.817*
For Aesthetic reasons	17.2	152€	27.8	1.30	*0.243*
To support the community of the game	17.4	55€	34.8	2.04	*<0.001*
Other reasons	1.6	21€	21.0	1.63	*0.503*
**Spending** 1€ or less/month	17.8		8.0	ref.	ref.
]1–5€[	6.0		10.7	1.62	0.364
[5–10€[	11.2		10.1	1.28	0.585
[10–20€[	15.9		14.0	1.74	0.158
[20–50€]	30.8		28.8	4.07	<0.001
more than 50€/month	18.3		39.9	6.79	<0.001

* adjusted OR for all variables; for reasons for payment items, no is the reference.

## Data Availability

Not applicable.

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
