# Peer review of "Spending Money in Free-to-Play Games: Sociodemographic Characteristics, Motives, Impulsivity and Internet Gaming Disorder Specificities"

_ijerph, 2022, doi:10.3390/ijerph192315709_

Round 1

Reviewer 1 Report

This could be a topic that generates interests to researchers in the gaming area. My only comment to the authors are that:

There seems to be many variables involved in each multivariate analysis, multicollinearity needs to be calculated, since score for each dimension of a given scale instead of the total score was used in analysis.

Author Response

We thank the reviewer for his positive appreciation of our work and his relevant remark.

·       My only comment to the authors are that: There seems to be many variables involved in each multivariate analysis, multicollinearity needs to be calculated, since score for each dimension of a given scale instead of the total score was used in analysis.

We agree with the reviewer that, in the presence of multicollinearity, regression models become unstable. We assessed multicollinearity by computing a variance inflation factor score, which measures how much the variance of a regression coefficient is inflated due to multicollinearity in the model. For Gaming patterns, Gaming experience and Gaming motives variables, a preliminary analysis on the correlations was carried out. The correlations between modalities for Gaming motives being strong, the analysis of VIFs led us to remove two items from the motivation variable: “Social” and “Fantasy”. This point has been clarified in the section “2. Materials and Methods”.

Reviewer 2 Report

A very well written and researched paper. It sets up the study very clearly, uses a large sample with appropriate measures and provides a set of findings which are useful for the field. The psychological factors associated with micro-transaction expenditure have been under-researched.

My only small criticism of the paper is that it contains a lot of measures: it's bit a a kitchen-sink of variables, but the author have justified the variables.

-I'd maybe include more info on the scoring range for measures in the Method section.

-Need to make the % of IGD cases a bit clearer in the write-up. The Pontes and Griffiths measure is a potential limitation. It tends to yield very high estimates.

-In some places in the Discussion the authors go beyond their own findings and splice in findings from other studies to justify a conclusion which is not based on their own data, e.g., playing longer = harm on gamers' lives and the reference to mediation effects that were not tested here.

-Placing spending limits on people's gaming sounds like an affordability test and the actions of authoritarian governments that shall not be named. I'd be more cautious about this type of statement and whether govts should ever be telling people how much they spend on a given class of activity.

Author Response

We thank the reviewer for his positive appreciation of our work and his interesting comments and relevant remarks.

·       I'd maybe include more info on the scoring range for measures in the Method section.

Addition have been made in text (part 2.2. Measures

·       Need to make the % of IGD cases a bit clearer in the write-up. The Pontes and Griffiths measure is a potential limitation. It tends to yield very high estimates.

Addition have been made in text (part 3.3. Internet gaming disorder)

·       Placing spending limits on people's gaming sounds like an affordability test and the actions of authoritarian governments that shall not be named. I'd be more cautious about this type of statement and whether govts should ever be telling people how much they spend on a given class of activity.

We modified the conclusion accordingly. 

Reviewer 3 Report

In this paper the authors utilized a wide scale Internet based survey conducted in France in 2019 in which 5062 participants answered self-administered questioneers regarding their gaming experience, sociodemographic characteristics, gaming experience, impulsivity and others. From that total sample, 3472 were gamers and their answers were analyzed and reported in this paper.

The main motivation, as stated by the authors, was to explore gamers who spend money in games to those who do not spend money. With that in mind, the authors present various correlates such as that men tend to play more than women, that older people (50 and above) play less, that people who recently started playing (first two years) are more likely to spend money, and so on.

The methodology of the study looks sound and valid. The paper itself is clearly written and the logical flow of ideas is well administrated, as well as the related work information. My main concern is with respect to the significance of the work. Most of the reported results seems to be quit obvious to me. I couldn't find any surprising insight from the presented analysis (e.g. people who spend money of games are more susceptible to IGD seems pretty obvious). So one question that comes to mind is: what are the significant contributions of the work? How does it advance the current scientific knowledge? And what are the *practical* implications of the results?

To answer these questions, I suggest the authors to add new text to discuss the implications of the study. Currently, while the discussion section is indeed long, it fails to explain the overall picture of the study. Instead of dealing with each factor individually, I think that part of the discussion should be in a more general term. In other words, please try to paint a picture of the forest, and not just tackle each individual tree. For example, are there any implications for public health officials? Regulators? Maybe to the gaming industry?

(page 6, first sentence of 3.2 "was more very significantly more"…)

Author Response

We thank the reviewer for his positive appreciation of our work and his relevant comment.

·       Most of the reported results seems to be quite obvious to me. I couldn't find any surprising insight from the presented analysis (e.g. people who spend money of games are more susceptible to IGD seems pretty obvious). So one question that comes to mind is: what are the significant contributions of the work? How does it advance the current scientific knowledge? And what are the *practical* implications of the results? To answer these questions, I suggest the authors to add new text to discuss the implications of the study. Currently, while the discussion section is indeed long, it fails to explain the overall picture of the study. Instead of dealing with each factor individually, I think that part of the discussion should be in a more general term. In other words, please try to paint a picture of the forest, and not just tackle each individual tree. For example, are there any implications for public health officials? Regulators? Maybe to the gaming industry?

The existence of a link between spending money on game and IGD seems to be a reasonable hypothesis but had not yet been clearly established. Indeed, most of the studies that have studied this subject have focused on a very particular form of spending, the practice of loot-boxes. As far as we know, only one study has established and measured a link between spending money in free-to-play games and IGD, but this one was based on a sample of young people (12-18 years old). Our results allow us to consolidate the existence of this link

At the Reviewer's request, we have strengthened and clarified the conclusion by structuring it into three paragraphs that better address the three questions: what are the significant contributions of the work? How does it advance the current scientific knowledge? And what are the *practical* implications of the results?

·       (page 6, first sentence of 3.2 "was more very significantly more"…)

Correction have been done